# Magnitude and determinants of improved household latrine utilization in Ethiopia: Multilevel analysis of the mini Ethiopian Demographic Health Survey (EDHS) 2019

Aragaw Tesfaw[1]*, Mulu Tiruneh[1], Melkalem Mamuye[1], Zebader Walle[1], Wondossen Teshager[1], Fentaw Teshome[1], Alebachew Taye[2], Wondimnew Dessalegn[1], Gashaw Walle[3], Asaye Alemneh Gebeyehu[1]

1 Department of Public Health, College of Health Sciences, Debre Tabor University, Debre Tabor, Ethiopia,
2 Department of Statistics, College of Natural Science, Debre Tabor University, Debre Tabor, Ethiopia,
3 Department of Biomedical Science, College of Health Sciences, Debre Tabor University, Debre Tabor, Ethiopia

* aragetesfa05@gmail.com

**Data Availability Statement:** All relevant data are within the manuscript.

## Abstract

### Introduction

Lack of sanitation is a major global problem mainly for the poor and disadvantaged. According to the 2016 Ethiopian Demographic and Health Surveys (EDHS) report, one out of every three households lack a toilet in Ethiopia and about 56% of rural households use unimproved toilet facilities. We aimed to determine the magnitude of improved household latrine utilization and its determinants in Ethiopia using the mini–Ethiopian Demographic Health Survey (EDHS) 2019 data set.

### Method

A secondary data analysis was conducted based on the mini 2019 EDHS data set. A total weighted sample of 8663 households were involved in analysis. After selecting the relevant variables for the outcome variable, we have fitted four different models. The null (empty) model with no independent variables and the second model contained the effects of the individual-level factors on the outcome variable. The third model included the influence of the community-level factors on the response variable, and the final multilevel multivariable logistic regression model examined the effects of individual-level and community-level variables on the outcome variable. The measure of variation was quantified using Intra-Class Correlation (ICC), Median Odds Ratio, and Proportional Change in Variance (PCV). The Adjusted Odds Ratio (AOR) with a 95% Confidence Interval (CI) was used to show the strength of association and statistical significance was declared at p value < 0.05.

### Results

The magnitude of improved latrine utilization in Ethiopia was 19.5% with 95% CI (18.6%, 20.3%). The factors: educational status (AOR = 1.67; 95% CI: (1.10, 2.55), highest wealth

**Funding:** The author(s) received no specific funding for this work.

**Competing interests:** The authors have declared that no competing interests exist.

**Abbreviations:** DHS-, Demographic Health Survey; SNNPR, Southern Nations Nationalities and People Region; WHO, World Health Organization.

index (AOR = 3.73; 95% CI: (2.73, 5.12), urban residence (AOR = 3.09; 95% CI: (1.68, 5.67), living in Addis Ababa (AOR = 4.08; 95% CI: (1.03, 16.2) and Dire Dawa (AOR = 8.22; 95% CI: (2.46, 27.42) and Somali regions (AOR = 3.11; 95% CI: (1.15, 8.42) were significantly associated to improved latrine utilization in Ethiopia.

## Conclusion

The magnitude of improved latrine utilization was quite low in Ethiopia. Higher wealth index, living in more urbanized areas, and the household head's educational status were all significant predictors of improved latrine utilization. The finding implies a need to increase household's access to latrine facilities and improve latrine utilization, particularly for rural households in the country.

## Background

Sanitation is the provision of facilities for the safe disposal of human excreta [1]. Unimproved sanitation facilities are responsible for increased risk mainly in low resource settings [2–4].

The provision of appropriate latrine facilities is essential for dignity, safety, health, and well-being [3]. Latrine utilization is the practice of regularly using existing latrines for safe disposal of excreta [5]. The total global economic losses associated with inadequate water supply and sanitation is high [6]. The total global economic losses associated with inadequate water supply and sanitation are high. Globally, 2 billion people lack basic sanitation services. More than 1.9 million deaths and 123 million disability-adjusted life-years (DALYs) could have been prevented by the provision of adequate access to water, hygiene and sanitation (WASH) [4, 7].

Worldwide, lack of sanitation is a serious health risk, affecting billions of people around the world, particularly the poor and disadvantaged. In the African Region, more than 500 million people lacked improved sanitation, and more than 231 million of them used open defecation. Only 30% of Sub-Saharan Africans used improved latrine facilities [8, 9]. Poor household latrine utilization increases the risk of transmission of diseases and it is responsible for about 30% of annual diarrheal deaths in low and middle-income countries, mainly in under-5 children [1, 10].

The issue of adequate WASH is a regional problem in countries around the world. It is a major problem in low-and middle-income countries [11, 12]. In developing countries, 47% of the population lives in an unhygienic environment, while in developed countries the proportion is only 1% [13]. Sub-Saharan Africa remained the most lagging in terms of accelerating access to improved latrine facilities. According to regional estimates, only 30% of Sub-Saharan Africans used improved latrine facilities [14]. According to World Health Organization (WHO) and United Nations Children's Fund (UNICEF) Joint monitoring program 2021 reports, 494 million people practice open defecation. Most of (92%) these people lived in rural areas and nearly half of them lived in sub-Saharan Africa [15]. In Ethiopia, only 51.5% of health care facilities used improved sanitation facilities [16] and 60% of the communicable disease burden is related to poor WASH, and more than 250,000 children die every year from WASH-related diseases in the country [6, 17]. According to the 2016 EDHS report, 56% of the rural households use unimproved toilet facilities [9]. Similarly, a systematic review conducted in Ethiopia found a high prevalence of trachoma associated with the lack of improved latrine utilization [18]. Several studies in Ethiopia showed variations on utilization of improved

latrine facilities in different parts of the country. A study conducted in Southwest Ethiopia found that only 25.36% of households used improved latrines [4] while a study conducted in rural households of Northwest Ethiopia indicated that the proportion of utilization of improved latrine was 49.1% [3]. According to the 2016 EDHS report, more than half of households, 53% (43% urban and 56% rural), use unimproved latrine facilities, with 32% (7% urban and 39% rural) still practicing open defecation [19]. While the 2019 EHDS report showed that overall, 20% of Ethiopian households use improved toilet facilities (42% in urban areas and 10% in rural areas. About (56%) of rural households use unimproved toilet facilities [20].

Several studies identified the factors associated with improved latrine utilization: Sex of household heads, region, residence, family size, age of household head, educational level, marital status of the household heads, and source of drinking water were some of the factors linked to improved latrine utilization [3–5, 8]. However, prior studies on the issue are focused on specific geographical regions in the country, moreover there are inconsistences between the study's findings. Therefore, there is a need to establish evidence-based information on the magnitude of improved latrine utilization status and the factors associated with it in the households of Ethiopia. Hence, an understanding of improved latrine utilization practices and determinant factors will guide to plan and develop targeted intervention programs for the improvement of household latrine utilization in the country which is included in the United Nations Sustainable Development Goals (SDGs) (Goal 6) with the target of achieving access to basic sanitation for all [21]. Moreover, the findings of this study will be used to generate further and stronger information that can complement the original survey report.

## Methods

### Study setting, design, and period

This study was a secondary data analysis from the 2019 mini-Ethiopian Demographic Health Survey (MEDHS) data set which was collected from nine regional states and two administrative cities (Addis Ababa and Dire-Dawa) of urban and rural areas of Ethiopia. The 2019 mini-EMDHS was a cross-sectional survey conducted from 21 March to 28 June 2019 at the national level. Ethiopia is one of the most popular countries in the Horn of Africa with great geographical diversity and an estimated population of more than 112 million [22]. The country comprises nine regions and two administrative cities; Tigray, Afar, Amhara, Oromia, Somali, Benshangul-Gumuz, Southern Nations Nationalities and People (SNNP), Gambela, and Harari. Region is administratively divided into zones, zones into woredas, and each woreda is subdivided into the smallest units called kebele. According to the 2019 Ethiopian Population and Housing Census (EPHC), a complete list of 149,093 enumeration areas (EAs) were used as the sampling frame for the 2019 EMDHS [23].

### Data sources, study population, and sampling

The data set for this study was obtained from the 2019 Ethiopian Mini Demographic and Health Survey (EMDHS) which was conducted by the Central Statistics Agency (CSA) with International Conference Federation. EMDHS is a population-based household survey designed to provide nationally representative data for the entire country and to estimate key demographic and health indicators. The sample for the 2019 EMDHS was designed to provide estimates of key indicators for the country as a whole, for urban and rural areas separately, and for each of the nine regions and the two administrative cities [24]. The survey used a household questionnaire to identify eligible household heads for interviews and to collect information on socio-demographic and other health-related indicators including hygiene and sanitation. In this analysis, a total of 8663 households were eligible for interview. The survey

used a two-stage stratified cluster sampling to select individual samples (households). In the first stage, a total of 305 EAs (93 urban and 212 rural) were selected proportional to EA size. In the second stage, from the listed sampling frame, an average of 30 households were selected per EA with equal probability. In our analysis, we used the HR data set and the study population were household members who have used latrines in the past five years of the survey. Therefore we include all eligible samples (households) who were participated in the 2019 mini–Ethiopian Demographic Health Survey based on predefined selection criteria (study variables and DHS guide line) related to the specific objectives of our study. A weighted sample of 8663 individuals were involved in the analysis. Any further information about methodology or sampling procedures is presented in the 2019 EMDHS report and guide [23, 24].

## Study variables

The outcome variable was improved latrine utilization, while the independent variables were categorized into individual level and community level factors. The individual level factors were age, sex, educational status, household wealth, exposure to mass media, and location of the water source. The two variables place of residence and region were considered as community level factors.

## Definition of terms

- **Latrine utilization** is defined as the use of the latrine by all family members (above age 5 years) in the households [8].

- **Improved sanitation facilities**: Households who had a private improved pit latrine with a slab or vented improved pit latrine or composting toilet, flush or pour/flush facility connected to a piped sewer system/ septic tank/pit, regardless of whether it is shared with other households [25].

- **Improved latrine Utilization**: Households were considered as properly utilizing their latrines if the latrine is hygienic and every member of the household whose age is above 5 are reported to use the facility by the respondents, there is safe disposal of child feces, no observable feces in the compound and /or latrine slab, at least one sign of use (clear footpath to the latrine not covered by grass or anything, the latrine is smelly, presence of anal cleansing material, or the slab is wet [15].

## Data management and Statistical analysis

The data were extracted, recoded, and further analysis was done using Stata version 17 software. In the analysis, sampling weight was used to adjust the non-proportional allocation of the sample to strata and regions during the survey process and to restore the representativeness of the data. Descriptive statistics like weighted frequencies and other summary statistics were used to describe and present the study population. First, we did a bivariable multilevel logistic regression analysis, and those variables with p values less than 0.25 were included in the multivariable multilevel logistic regression analysis. A multivariable multilevel logistic analysis was used to identify statistically significant factors associated with improved latrine utilization.

After selecting the relevant variables for the outcome variable, we have fitted four different models. The first model was the null model, which is an empty model developed without individual or community level variables, and the second model (Model II) contained the effects of the individual-level variables on the response variable. The third model (Model III) included

the influence of the community-level variables on the response variable, and the final model examined the effects of individual-level and community-level variables.

The measure of variation in improved latrine utilization was quantified using Intra-Class Correlation (ICC), Median Odds Ratio (MOR), and Proportional Change in Variance (PCV). ICC measures the degree of similarity within a group or cluster. If ICC value is greater than 10, we did a multilevel logistic analysis than the standard logistic analysis to account for the hierarchical nature of data. A MOR is the median value of a set of odds ratios between an individual at the higher risk cluster and at the lower risk clusters when randomly picked out of two clusters. It measures the degree of variation of improved latrine utilization practice across clusters. If the MOR value is greater than one, it shows the presence of variation across clusters [26]. Proportional change in variance (PCV) also measures the explained difference in variance in the final model by combining both individual and community level factors [27].

Among four fitted models, the better model was chosen using deviance statistics, since it measures model fitness. A model with a lower deviance value was considered better-fitted model for this study. Multicollinearity among independent variables was checked using the variance inflation factor (VIF). Adjusted odds ratio with 95% confidence interval was used for measuring the strength of association between independent variables with improved latrine utilization.

## Ethics approval and consent

Data set for this study was obtained from the DHS program. We have accessed and downloaded the data set after describing the purpose of the study in the online platform. The data sets obtained from the DHS program were kept confidential and excluded the participant's personal identifiers. The data set is publicly available with reasonable request from the DHS program at https://dhsprogram.com/Data/terms-of-use.cfm. The research is conducted according to the Declaration of Helsinki.

## Results

### Socio-demographic and economic characteristics of participants

A total of 8663 households (weighted) were included in the analysis. Among these, about 6,751 (77.93%) of the participants were males and 1912 (22.07%) were females. The mean age of the respondents was 43 with SD (±16 years). The majority 3,207 (37.02%) of the respondents were Oromo by ethnicity, followed by Amhara 585 (6.75%). About 2664 (30.75%) of the respondents were Urban and 5,999 (69.25%) of the respondents were rural by place of residence. Regarding to education status; about 4,120 (47.56%) of the survey participants did not have education, 601 (6.94%) of them were in higher education level. Concerning to Economical states, i.e., wealth index category; about 3,133 (36.16%) of the study subjects were poor, 1,675 (19.34%) of the study subjects were in the middle wealth index category, and 3,855 (44.50%) of the study subjects were Rich (Table 1).

### Latrine coverage and type of latrine facilities in Ethiopia

The following table showed that 5,818 (72.91%) of the subjects had a latrine facility. Among these, only 1,685 (19.46%) of households had improved latrines. About 4,302 (68.11%) of the households share a toilet with other households. In 205 (3.24%) of the households, the location of the toilet facility is in the own dwelling. The most common type of latrine used by the households is pit latrine without slab/open pit. About 2347(27.09%) of the households did not have a latrine facility rather they used bush or open fields for defecation (Table 2).

**Table 1. Socio-demographic and economic characteristics of participants, EDHS 2019.**

| Variables | Category | Weighted Frequency | Weighted Percentage |
|---|---|---|---|
| Region | Tigray | 585 | 6.75 |
| | Afar | 88 | 1.01 |
| | Amhara | 2,110 | 24.35 |
| | Oromia | 3,207 | 37.02 |
| | Somali | 419 | 4.83 |
| | Benishangul- Gumuz | 94 | 1.08 |
| | SNNPR | 1,668 | 19.25 |
| | Gambela | 35 | 0.41 |
| | Harari | 25 | 0.29 |
| | Addis ababa | 376 | 4.34 |
| | Dire dawa | 57 | 0.66 |
| Residence | Urban | 2664 | 30.75 |
| | Rural | 5,999 | 69.25 |
| Sex | Male | 6,751 | 77.93 |
| | Female | 1912 | 22.07 |
| Educational status | no education | 4,120 | 47.56 |
| | primary | 3,069 | 35.43 |
| | secondary | 872 | 10.07 |
| | Higher | 601 | 6.94 |
| Age of respondent | <40 year | 4,497 | 51.91 |
| | 40–60 year | 2,760 | 31.87 |
| | 61–95 year | 1,338 | 15.44 |
| | Do not know | 68 | 0.78 |
| Wealth index | Poor | 3,133 | 36.16 |
| | Middle | 1,675 | 19.34 |
| | Rich | 3,855 | 44.50 |
| Family size | ≤5 | 5,516 | 63.68 |
| | >5 | 3147 | 36.32 |
| Exposure to media | Yes | 6,222 | 71.82 |
| | No | 2,441 | 28.18 |

**Table 2. Latrine coverage and related conditions in Ethiopia, EDHS 2019.**

| Variables | Category | Weighted Frequency | Weighted Percentage |
|---|---|---|---|
| Do you have latrine facility | Yes | 5,818 | 72.91 |
| | No | 2845 | 27.09 |
| Type of Latrine Facility | Improved | 1,685 | 19.46 |
| | Unimproved | 6,978 | 80.54 |
| share toilet with other households | Yes | 4,302 | 68.11 |
| | No | 2,014 | 31.89 |
| Number of households member sharing toilet | <5 household members | 1259 | 62.5 |
| | ≥5 household members | 737 | 36.59 |
| | Don't know | 16 | 0.79 |
| Location of Toilet facility | In own dwelling | 205 | 3.24 |
| | In own yard/plot | 5,461 | 86.46 |
| | Elsewhere | 651 | 10.30 |

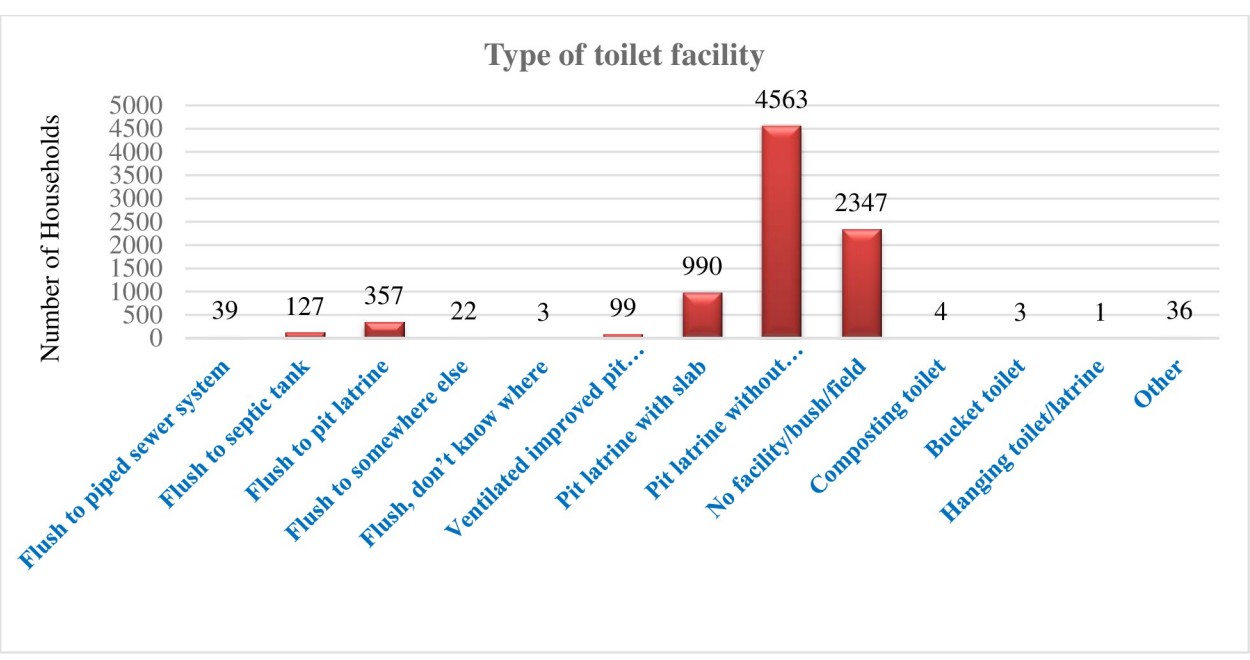

**Fig 1. Type of toilet facilities in Ethiopian households, EDHS, 2019.**

The most common type of latrine used by the households is pit latrine without slab/open pit. About 2347 (27.09%) of the households did not have latrine facility rather they used bush or open fields for defecation (Fig 1).

## Source of water supply and means of communication for the households

As the following table shows, the 2,655(30.65%) common source of drinking water for the households was public tap/standpipe and the majority 6,901(97.82%) the location of their water source was elsewhere outside their dwelling (Table 3).

## Latrine utilization practice in Ethiopian households

As the following table shows, a large proportion of households 561(65.5%) whose household head were educated, with higher education or more were used improved latrine as compared with those households headed by uneducated household heads. Those households who have in the highest wealth index category used improved latrine than those households with poor wealth index category, 207(5.9%). The majority 760(32.0%) of female headed households used improved latrine than male headed households 1554(24.7%). The majority of households in Addis Ababa and Dire Dawa used improved latrines than other regions of Ethiopia (Table 4).

## Determinate of improved latrine utilization in Ethiopia

In the multilevel multivariable analysis, the final fitted logistic regression model showed that educational status of the head of household, wealth index, exposure to media, urban residence, living in Addis Ababa, Dire Dawa and Somali regions were significantly associated factors for improved latrine utilization in Ethiopia. Those household heads who have a higher education level were approximately 2 times more likely to utilize improved latrine utilization than those who have no education (AOR = **1.67;** 95% CI: (1.10,2.55). Households with rich wealth index category were more likely improved latrine than the poor wealth index group households

**Table 3. Source of water supply and means of communication for the household, EDHS, 2019.**

| Variables | Category | Weighted Frequency | Weighted Percentage |
|---|---|---|---|
| Source of drinking water | piped into dwelling | 193 | 2.23 |
| | piped to yard/plot | 1,010 | 11.66 |
| | piped to neighbor | 326 | 3.77 |
| | public tap/standpipe | 2,655 | 30.65 |
| | tube well or borehole | 339 | 3.92 |
| | protected well | 423 | 4.89 |
| | unprotected well | 402 | 4.65 |
| | protected spring | 849 | 9.81 |
| | unprotected spring | 1,183 | 13.66 |
| | river/dam/lake/ponds/stream/canal/ | 1,097 | 12.66 |
| | Rainwater | 35 | 0.40 |
| | tanker truck | 39 | 0.45 |
| | cart with small tan | 5 | 0.05 |
| | bottled water | 78 | 0.90 |
| | Other | 27 | 0.31 |
| location of source for water | In own dwelling | 10 | 0.14 |
| | In own yard/plot | 144 | 2.05 |
| | Elsewhere | 6,901 | 97.82 |

(AOR = 3.73; 95% CI: (2.73, 5.12). The odds of using improved latrine was higher among households residing in urban areas than living in rural areas (AOR = 3.09; 95% CI: (1.68, 5.67). People living in big cities ((Addis Ababa (AOR = 4.08; 95% CI: (1.03, 16.2) and Dire Dawa (AOR = 8.22; 95% CI: (2.46, 27.42)) of the country were more likely to have improved latrine than their counter parts. The odds of having improved latrine utilization were approximately three times higher in Somalia region than other regions of the country (AOR = 3.11; 95%CI: (1.15, 8.42). The odds of using improved latrine was higher among households exposed to media than households who did not have access to media (AOR = 1.39; 95% CI: (1.08, 1.8) (Table 5).

## The variation of the fitted models after multilevel analysis

As the table shows, the cluster level variance is lower in the final model. The intra-cluster correlation coefficient is reduced from 75% to 36% in the final model. The median odds ratio is also decreased from 19.72 to 3.63 in the final model (Table 6).

## Discussion

Utilization of improved sanitation facilities used to decrease several infectious disease, mainly diarrheal diseases and helminthic infections. Therefore, achieving universal access to better sanitation facilities to all households is crucial [1, 28]. Lack of sanitation is a global challenge mainly for the disadvantaged and underprivileged groups, resulting poor school attendance, anxiety, stunting and reduced cognitive performance [12, 29]. Our study determined the magnitude of improved household latrine utilization and its determinants in Ethiopia using MEDHS data set. According to our study, 19.46% of households had improved latrine facilities. This result is lower than findings from low-income informal settlers in East African cities [12]. This variation could be due to differences in the study setting, sample population, and socio-economic development of the countries.

**Table 4. Latrine utilization practice in Ethiopian households, mini EDHS, 2019.**

| Variable | Category | Unimproved latrine utilization | Improved latrine utilization |
|---|---|---|---|
| Sex | Male | 4737(75.3%) | 1554(24.7%) |
| | Female | 1612(68.0%) | 760(32.0%) |
| Educational status | No-education | 3500(84.8%) | 628(15.2%) |
| | Primary | 2049(75.5%) | 666(24.5%) |
| | Secondary | 504(52.3%) | 459(47.7%) |
| | Higher | 296(34.5%) | 561(65.5%) |
| Age | 15–30 year | 1776(70.5%) | 744(29.5%) |
| | 31–40 year | 1682(73.5%) | 605(26.5%) |
| | ≥41–50 year | 1085(75.2%) | 358(24.8%) |
| | 51–60 | 889(75.2%) | 293(24.8%) |
| | >60 | 917(74.5%) | 314(25.5%) |
| Wealth index | Poor | 3291(94.1%) | 207(5.9%) |
| | Middle | 1163(90.5%) | 122(9.5%) |
| | Rich | 1895(48.8%) | 1985(51.2%) |
| Exposure to media | Yes | 4159(66.0%) | 2139(34.0%) |
| | No | 2190(92.6%) | 175(7.4%) |
| Location for source for water | In own dwelling | 9(75.0%) | 3(25.0%) |
| | In own yard/plot | 105(80.2%) | 26(19.8%) |
| | Elsewhere | 5671(89.0%) | 703(11.0%) |
| Residence | Urban | 917(34.7%) | 1728(65.3%) |
| | Rural | 5432(90.3%) | 586(9.7%) |
| Region | Tigray | 553(77.5%) | 161(22.5%) |
| | Afar | 567(85.4%) | 97(14.6%) |
| | Amhara | 842(83.6%) | 165(16.4%) |
| | Oromia | 912(89.6%) | 106(10.4%) |
| | Somali | 522(79.5%) | 135(20.5%) |
| | Benishangul- Gumuz | 671(91.4%) | 63(8.6%) |
| | SNNPR | 923(90.8%) | 94(9.2%) |
| | Gambela | 631(91.1%) | 62(8.9%) |
| | Harari | 353(49.1%) | 366(50.9%) |
| | Addis Ababa | 144(20.5%) | 558(79.5%) |
| | Dire Dawa | 231(31.3%) | 507(68.7%) |

In the multilevel analysis, the final fitted logistic regression model showed that educational status of the head of household, wealth index, urban residence, living in Addis Ababa, Dire Dawa and Somali regions were significantly associated factors for improved latrine utilization in Ethiopia. Higher educated household heads are roughly twice more likely to have improved latrine utilization practices than uneducated heads of households. This finding is similar to other studies in which households with educated heads are more likely to utilize improved latrines than households without educated heads [11, 25, 30–32]. This could be explained as education is an important factor for making more informed decisions about health and for wise use of resources to build and utilize improved latrine facilities. Households in the rich wealth index category were 3.7 times more likely to use improved latrines than the poor wealth index category. This is because the number of options and opportunities will be increased as household wealth increases. So that households with a stable economy can be able to afford the materials needed to install improved latrine facilities of higher quality than those with poor wealth index category.

**Table 5. Multilevel multivariable logistic regression analysis of determinants of improved latrine utilization using individual and community level factors, EDHS 2019.**

| Variable | Category | latrine utilization | | Model 2 (AOR, 95%CI) | Model 3 (AOR, 95%CI) | (Final model) (AOR, 95%CI) |
|---|---|---|---|---|---|---|
| | | Unimproved | Improved | | | |
| Sex | Male | 4737(75.3%) | 1554(24.7%) | 1 | | 1 |
| | Female | 1612(68.0%) | 760(32.0%) | 0.918(0.73,1.16 | | 0.88(0.69,1.11) |
| Educational status | No-education | 3500(84.8%) | 628(15.2%) | 1 | | 1 |
| | Primary | 2049(75.5%) | 666(24.5%) | 0.99(0.78,1.27) | | 1.03(0.81,1.31) |
| | Secondary | 504(52.3%) | 459(47.7%) | 1.005(0.69,1.46) | | 1.01(.69, 1.47) |
| | Higher | 296(34.5%) | 561(65.5%) | 1.54(1.01,2.39) | | **1.67(1.10,2.55)** |
| Age | 15–30 year | 1776(70.5%) | 744(29.5%) | 1 | | 1 |
| | 31–40 year | 1682(73.5%) | 605(26.5%) | 0.94(0.73, 1.22) | | 0.97(0.75,1.25) |
| | ≥41–50 year | 1085(75.2%) | 358(24.8%) | 1.11(0.82,1.51) | | 1.14(0.82,1.54) |
| | 51–60 | 889(75.2%) | 293(24.8%) | 1.10(0.79,1.53) | | 1.13(0.814,1.58) |
| | >60 | 917(74.5%) | 314(25.5%) | 1.25(0.89, 1.77) | | 1.33(0.95,1.88) |
| Wealth index | Poor | 3291(94.1%) | 207(5.9%) | 1 | | 1 |
| | Middle | 1163(90.5%) | 122(9.5%) | 1.92(1.41,2.59) | | **1.99(1.46,2.71)** |
| | Rich | 1895(48.8%) | 1985(51.2%) | 4.08(3.01,5.53) | | **3.73(2.73, 5.12)** |
| Exposure to media | Yes | 4159(66.0%) | 2139(34.0%) | 1.39(1.09,1.8) | | **1.39(1.08,1.8)** |
| | No | 2190(92.6%) | 175(7.4%) | 1 | | 1 |
| Location for source for water | In own dwelling | 9(75.0%) | 3(25.0%) | 1 | | 1 |
| | In own yard/plot | 105(80.2%) | 26(19.8%) | 0.31(0.04,2.49) | | 0.32(0.05,2.28) |
| | Elsewhere | 5671(89.0%) | 703(11.0%) | 0.19(0.03, 1.41) | | 0.21(0.32,1.35) |
| Residence | Urban | 917(34.7%) | 1728(65.3%) | | 21.28(12.7,35.5) | **3.09(1.68, 5.67)** |
| | Rural | 5432(90.3%) | 586(9.7%) | | 1 | |
| Region | Tigray | 553(77.5%) | 161(22.5%) | | 1 | 1 |
| | Afar | 567(85.4%) | 97(14.6%) | | 0.42(0.15,1.13) | 1.13(0.39,3.3) |
| | Amhara | 842(83.6%) | 165(16.4%) | | 1.03(0.42,2.52) | 1.96(0.79, 4.87) |
| | Oromia | 912(89.6%) | 106(10.4%) | | 0.288(0.11,0.73) | 0.422(0.158,1.13 |
| | Somali | 522(79.5%) | 135(20.5%) | | 0.78(0.29,2.09) | **3.11(1.15,8.42)** |
| | B/Gumuz | 671(91.4%) | 63(8.6%) | | 0.35(0.13,0.98) | 0.615(0.22,1.71) |
| | SNNPR | 923(90.8%) | 94(9.2%) | | 0.52(0.21,1.3) | 0.889(0.348,2.27) |
| | Gambela | 631(91.1%) | 62(8.9%) | | 0.207(0.07,0.59) | 0.35(0.12,1.04) |
| | Harari | 353(49.1%) | 366(50.9%) | | 2.06(0.79,5.39) | 2.38(0.82, 6.89) |
| | Addis Ababa | 144(20.5%) | 558(79.5%) | | 5.86 (2.07,16.6) | **4.08(1.03,16.2)** |
| | Dire Dawa | 231(31.3%) | 507(68.7%) | | 7.29(2.75,19.3) | **8.22(2.46,27.42)** |

Key: Bold indicates statistically significant findings

**Table 6. The variation of the fitted models after multilevel analysis, EDHS, 2019.**

| Random variation | Null model | Model I | Model II | Final Model |
|---|---|---|---|---|
| Cluster level variance | 9.85 | 2.99 | 2.39 | 1.84 |
| ICC | 0.75 | 0.478 | 0 .42 | 0.36 |
| MOR | 19.72 | 5.2 | 4.34 | 3.63 |
| PCV (%) | Reference | 69.64 | 75.74 | 81.32 |
| Deviance | 5,888.8 | 3,580.5 | 5,607.97 | 3,496.56 |

Key: ICC: Intra Correlation Coefficient, MOD: Median Odds Ratio, PCV: Proportional Change in Variance

The odds of improved latrine utilization were higher among households residing in urban areas than living in rural areas. This finding is consistent with other study reports [5, 9, 13, 30]. This could be attributed to the majority of people living in rural areas used more water for agriculture than for sanitation. Additionally, financial issues, lack of access to information can be the possible reasons for low utilization of improved latrine facilities in rural households.

Households residing in Addis Ababa & Dire Dawa were more likely to use improved latrines than their counterparts. This might be because these are the big administrative cities in the country and the households in these cities are more likely to have access to improved latrine facility as compared to other regions in the country. In addition, people living in urban areas are more likely to have access to information than people living in rural areas. Similarly, utilization of improved latrine facilities was higher among households exposed to media than households who did not have access to media. Since exposure to different sources of media will increase the awareness of people on the importance of using improved latrine facilities. The finding is in line with other studies [33–35].

Our study also revealed that the probability of having improved latrine utilization were approximately 3 times higher in the Somalia region than in other regions of the country. This might be a cultural and religious differences in the region, which gives much focus for sanitation services. The finding is inconsistent with EDHS 2016 [2], probably this might be due to difference in the source of data utilized. Although improved sanitation is essential for the improvement of human health and economic growth [1] this study revealed that the utilization of improved sanitation is very low in Ethiopia. which is similar to other countries residing in Sub-Saharan Africa [12, 31]. As diarrheal diseases are the most common in Ethiopia [36] due to lack of improved sanitation and hygiene, the study implies a need to ensure adequate and equitable access to improved sanitation facilities in the country. Though our study identified important predictor factors for improved latrine unitization, the following factors were not statistically significant age, sex and family size as they were reported as significant factors in other studies. This might be due to different in sample size, variation on the methods used. Probably our study used a national representative population based data while others studies are conducted only in specific areas of the country. Therefore future researchers will consider this discrepancy.

## Strengths and limitations

This is a multilevel analysis which provides weighted evidence on the magnitude and determinant factors of improved latrine facilities from Mini EDHS 2019 data set. However, our findings have limitations since the mini-EDHS data set did not have all possible predictor variables for improved latrine utilization within the country.

## Conclusion and recommendation

The magnitude of improved household latrine utilization is quite low in Ethiopia. Educational status of the household head, higher wealth index, exposure to media, and living in more urbanized areas were important predictors for improved latrine utilization in the country. Our study also revealed that the availability of improved sanitation facilities and utilization is lower than the National Hygiene and Environmental Health Strategy's goals. The study suggests that there is a need to increase access to latrine facilities and use of improved household latrine utilization in Ethiopia. We recommend further research to be done using Multicounty level survey data i.e. by including Demographic health survey data's of several countries for making comparisons.

## Author Contributions

**Conceptualization:** Aragaw Tesfaw, Wondossen Teshager, Fentaw Teshome, Alebachew Taye, Wondimnew Dessalegn.

**Data curation:** Aragaw Tesfaw, Wondossen Teshager, Gashaw Walle.

**Formal analysis:** Aragaw Tesfaw, Melkalem Mamuye, Asaye Alemneh Gebeyehu.

**Investigation:** Aragaw Tesfaw, Melkalem Mamuye, Zebader Walle.

**Methodology:** Aragaw Tesfaw, Zebader Walle, Wondossen Teshager, Asaye Alemneh Gebeyehu.

**Project administration:** Aragaw Tesfaw.

**Resources:** Aragaw Tesfaw.

**Software:** Melkalem Mamuye, Zebader Walle, Wondossen Teshager, Asaye Alemneh Gebeyehu.

**Supervision:** Mulu Tiruneh, Alebachew Taye.

**Validation:** Asaye Alemneh Gebeyehu.

**Writing – original draft:** Aragaw Tesfaw, Mulu Tiruneh, Zebader Walle, Wondossen Teshager, Fentaw Teshome, Alebachew Taye, Wondimnew Dessalegn, Gashaw Walle, Asaye Alemneh Gebeyehu.

**Writing – review & editing:** Aragaw Tesfaw, Mulu Tiruneh, Wondossen Teshager, Fentaw Teshome, Alebachew Taye, Wondimnew Dessalegn, Gashaw Walle, Asaye Alemneh Gebeyehu.

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
