## [Decision Letter · Decision Letter 0]

14 Feb 2023

PONE-D-22-34120Magnitude and Determinants of improved household latrine Utilization in Ethiopia: Multilevel Analysis of the Mini Ethiopian Demographic Health Survey (EDHS) 2019PLOS ONE

Dear Dr. Tesfaw,

Thank you for submitting your manuscript to PLOS ONE. After careful consideration, we feel that it has merit but does not fully meet PLOS ONE’s publication criteria as it currently stands. Therefore, we invite you to submit a revised version of the manuscript that addresses the points raised during the review process.

We look forward to receiving your revised manuscript.

Kind regards,

Gizat Almaw

Academic Editor

PLOS ONE

Journal Requirements:

- https://doi.org/10.1371/journal.pone.0241270

In your revision ensure you cite all your sources (including your own works), and quote or rephrase any duplicated text outside the methods section. Further consideration is dependent on these concerns being addressed.

4. Please amend the manuscript submission data (via Edit Submission) to include author Asaye Alemneh.

6. We note you have included a table to which you do not refer in the text of your manuscript. Please ensure that you refer to Table 4 and 5 in your text; if accepted, production will need this reference to link the reader to the Table.

Reviewers' comments:

Reviewer's Responses to Questions

**Comments to the Author**

1. Is the manuscript technically sound, and do the data support the conclusions?

Reviewer #1: Partly

2. Has the statistical analysis been performed appropriately and rigorously? 

Reviewer #1: No

3. Have the authors made all data underlying the findings in their manuscript fully available?

Reviewer #1: No

4. Is the manuscript presented in an intelligible fashion and written in standard English?

Reviewer #1: No

5. Review Comments to the Author

Reviewer #1: Thank you very much for the opportunity given for me to review the manuscript entitled "Magnitude and Determinants of improved household latrine Utilization in Ethiopia: Multilevel Analysis of the Mini Ethiopian Demographic Health Survey (EDHS) 2019."

As mentioned, the authors did a secondary data analysis that used to determine factors associated with an improved latrine Utilization based on the mini–Ethiopian Demographic Health Survey (EDHS) 2019 data set. Given that, I forwarded my comments that I believed which could help authors to improve the quality of their manuscript.

General comment:

Overall, despite the known limitation, this study may provide complementary evidence related to latrine Utilization in Ethiopia. I think, the manuscript need language revision and it would be good to refer similar research articles and or tools used for secondary data analysis and follow the writing styles of standard (peer reviewed) published articles. There is mix-up of words and or phrases that are commonly used to report primary studies. For example, phrases such as “This study was conducted in Ethiopia” may give wrong impression and it seems this study was designed to undertake independent study that collect primary research data. Hence, I suggest the authors should revise the document thinking as you are using a secondary data that was collected as part of the national survey.

Specific comments

#Introduction: Though the objective of the current study is stated in the abstract, the rationale and aim of the study is not clearly explained in the main document (introduction). Indeed, at the end of the last paragraph the author stated, “The findings of this study will help to increase utilization of improved latrines in the households of Ethiopia”. Can you please justify how? Rather, such type of studies often used to generate further and stronger information that can complement the original survey report.

Please look at the national EDHS report that was published in May 2021 and articulate the objectives of your study to fil gaps that is not addressed in the report-Perhaps it is still possible to develop testable hypothesis based on your interest of outcome and the main exposure variable. It would be also good to present the conceptual framework that can explain the theoretical and logical relationship of study variables included in this study.

Method: brief description of study settings, design and sampling procedures of the survey might be important but that should not be more detailed. Instead, you can cite the published report and explain the structure and composition of the main dataset related to the main outcome variable and related factor variables used for this study. Similarly, instead of describing in detail the general approaches (methods) used to conduct the survey, it is better to provide enough information on the techniques and procedures on how you extract the study variables and data used to achieve your main objective.

From my reading (both the manuscript and EDHS survey report), I understood that authors included all eligible samples (households) who were participated in the 2019 mini–Ethiopian Demographic Health Survey. That might be Ok to include all participants, but a systematic sampling (selection) of study units using predefined criteria would be preferred. Meaning, once the maximum power is achieved for the specific objective, the authors should further consider quality of survey data. Not the sample size but such type of dataset requires advanced and sound data handling and management - that is mandatory while using secondary data analysis.

Similarly, it is not the mathematical formula and or the general assumption (principles) rather authors should clearly explain their own practical analysis techniques and the respective outputs- that used to decide the next procedure.

I think, definitions of terms should precede data management and analysis- not at the end of research methods.

Results: like that of the method, the descriptive part of your result section reflected the setting and background characteristics of survey participant- it seems the original report. It is not focused as that was expected from secondary data analysis studies.

It is not clear how and where the X2-test and the corresponding p-vale presented in Table-1 were generated while only the frequency and percentage of one-dimensional variables are presented- which is not cross-tabulated.

In general, the descriptive findings and the whole results lack the standard criteria used for data visualization (presentation)- perhaps the results (Tabular presentation) reflect the methodological limitation and I felt authors should give more emphasis for data management. For instance, at least those important variables should be recoded or transformed using simple methods and then the reader could grasp the key findings related to this study.

It is still difficult to understand the findings generated from the multilevel analysis (Modelling). It is not supported with descriptive reports that can provide additional information- If not mistaken I did not see any descriptive findings or estimates that compare the observed frequency related with the main outcome (Improved Vs Unimproved). I suggest either to modify the descriptive findings or add columns to include the observed frequencies [n (%)] in the multilevel analysis (Table 5).

I think, authors should provide their response and explanation to clarify the above issues and then we can proceed to the next section.

6. PLOS authors have the option to publish the peer review history of their article (what does this mean?). If published, this will include your full peer review and any attached files.

Reviewer #1: No

---

## [Author Response · Author response to Decision Letter 0]

1 Apr 2023

Thank you very much for all the comments provided regarding our manuscript entitled “Magnitude and Determinants of improved household latrine Utilization in Ethiopia: Multilevel Analysis of the Mini Ethiopian Demographic Health Survey (EDHS) 2019” which are fully accepted and included in the revised version. I have accordingly made necessary revisions on the paper following the comments provided from the reviewer and editor. For your kind consideration, please find a point-by-point response to the comments and a submitted new revised version of the manuscript. All new changes have been highlighted in dark blue in the main document in order to facilitate review.

I hope that you will find the edits as per your expectation and I look forwards to hear from you

Please kindly find our point-by-point response for the comments provided. 

Response to reviewer comments 

Subject: Reviewer comment for Research Article PONE-D-22-34120

Thank you very much for the opportunity given for me to review the manuscript entitled "Magnitude and Determinants of improved household latrine Utilization in Ethiopia: Multilevel Analysis of the Mini Ethiopian Demographic Health Survey (EDHS) 2019."

As mentioned, the authors did a secondary data analysis that used to determine factors associated with an improved latrine Utilization based on the mini–Ethiopian Demographic Health Survey (EDHS) 2019 data set. Given that, I forwarded my comments that I believed which could help authors to improve the quality of their manuscript. 

General comment: 

Overall, despite the known limitation, this study may provide complementary evidence related to latrine Utilization in Ethiopia. I think, the manuscript need language revision and it would be good to refer similar research articles and or tools used for secondary data analysis and follow the writing styles of standard (peer reviewed) published articles. There is mix-up of words and or phrases that are commonly used to report primary studies. For example, phrases such as “This study was conducted in Ethiopia” may give wrong impression and it seems this study was designed to undertake independent study that collect primary research data. Hence, I suggest the authors should revise the document thinking as you are using a secondary data that was collected as part of the national survey. 

Author response: We would like to thank and appreciate the reviewer for providing important comments and suggestions. We have now modified the document based on the comments provided by the reviewer in the main document after reviewing some similar research articles in the area.

Specific comments 

#Introduction: Though the objective of the current study is stated in the abstract, the rationale and aim of the study is not clearly explained in the main document (introduction). Indeed, at the end of the last paragraph the author stated, “The findings of this study will help to increase utilization of improved latrines in the households of Ethiopia”. Can you please justify how? Rather, such type of studies often used to generate further and stronger information that can complement the original survey report. 

Please look at the national EDHS report that was published in May 2021 and articulate the objectives of your study to fil gaps that is not addressed in the report-Perhaps it is still possible to develop testable hypothesis based on your interest of outcome and the main exposure variable. It would be also good to present the conceptual framework that can explain the theoretical and logical relationship of study variables included in this study. 

Author response: We accepted the reviewer comments and suggestions and we have now corrected/ modified in the main document based on the comments

Method: brief description of study settings, design and sampling procedures of the survey might be important but that should not be more detailed. Instead, you can cite the published report and explain the structure and composition of the main dataset related to the main outcome variable and related factor variables used for this study. Similarly, instead of describing in detail the general approaches (methods) used to conduct the survey, it is better to provide enough information on the techniques and procedures on how you extract the study variables and data used to achieve your main objective. 

From my reading (both the manuscript and EDHS survey report), I understood that authors included all eligible samples (households) who were participated in the 2019 mini–Ethiopian Demographic Health Survey. That might be Ok to include all participants, but a systematic sampling (selection) of study units using predefined criteria would be preferred. Meaning, once the maximum power is achieved for the specific objective, the authors should further consider quality of survey data. Not the sample size but such type of dataset requires advanced and sound data handling and management - that is mandatory while using secondary data analysis. 

Similarly, it is not the mathematical formula and or the general assumption (principles) rather authors should clearly explain their own practical analysis techniques and the respective outputs- that used to decide the next procedure. 

I think, definitions of terms should precede data management and analysis- not at the end of research methods. 

Author response: We have corrected/ modified at the main document based on the comments

Results: like that of the method, the descriptive part of your result section reflected the setting and background characteristics of survey participant- it seems the original report. It is not focused as that was expected from secondary data analysis studies. 

It is not clear how and where the X2-test and the corresponding p-vale presented in Table-1 were generated while only the frequency and percentage of one-dimensional variables are presented- which is not cross-tabulated. 

In general, the descriptive findings and the whole results lack the standard criteria used for data visualization (presentation)- perhaps the results (Tabular presentation) reflect the methodological limitation and I felt authors should give more emphasis for data management. For instance, at least those important variables should be recoded or transformed using simple methods and then the reader could grasp the key findings related to this study. 

It is still difficult to understand the findings generated from the multilevel analysis (Modelling). It is not supported with descriptive reports that can provide additional information- If not mistaken I did not see any descriptive findings or estimates that compare the observed frequency related with the main outcome (Improved Vs Unimproved). I suggest either to modify the descriptive findings or add columns to include the observed frequencies [n (%)] in the multilevel analysis (Table 5). 

I think, authors should provide their response and explanation to clarify the above issues and then we can proceed to the next section. 

Author response: We have corrected/ modified at the main document based on the comments. We add descriptive statistical analysis results in the revised document based on the reviewer suggestions.

---

## [Editor Report · Decision Letter 1]

11 May 2023

PONE-D-22-34120R1Magnitude and Determinants of improved household latrine Utilization in Ethiopia: Multilevel Analysis of the Mini Ethiopian Demographic Health Survey (EDHS) 2019PLOS ONE

Dear Dr. Tesfaw,

Thank you for submitting your manuscript to PLOS ONE. After careful consideration, we feel that it has merit but does not fully meet PLOS ONE’s publication criteria as it currently stands. Therefore, we invite you to submit a revised version of the manuscript that addresses the points raised during the review process.

We look forward to receiving your revised manuscript.

Kind regards,

Gizat Almaw

Academic Editor

PLOS ONE
---

## [Author Response · Author response to Decision Letter 1]

25 May 2023

Thank you very much for all the comments provided regarding our manuscript entitled “Magnitude and Determinants of improved household latrine Utilization in Ethiopia: Multilevel Analysis of the Mini Ethiopian Demographic Health Survey (EDHS) 2019” which are fully accepted and included in the revised version. I have accordingly made necessary revisions on the paper following the comments provided from the reviewer and editor. For your kind consideration, please find a point-by-point response to the comments and a submitted new revised version of the manuscript. All new changes have been highlighted in dark blue in the main document in order to facilitate review.

I hope that you will find the edits as per your expectation and I look forwards to hear from you

Please kindly find our point-by-point response for the comments provided. 

Response to academic editor comments

Journal Requirements:

Author response: We prepared our manuscript using PLOS ONE's styles /templates 

- https://doi.org/10.1371/journal.pone.0241270

In your revision ensure you cite all your sources (including your own works), and quote or rephrase any duplicated text outside the methods section. Further consideration is dependent on these concerns being addressed.

Author response: We have now modified the document based on the comments provided by the editor in the main document 

Author response: The data for this study will be obtained from the cross-ponding author on a reasonable request

4. Please amend the manuscript submission data (via Edit Submission) to include author Asaye Alemneh.

Author response: We have included author Asaye Alemneh in the submission link based on the comments

Author response: We have now put the ethics statement in the methods section of the main document based on the comments

6. We note you have included a table to which you do not refer in the text of your manuscript. Please ensure that you refer to Table 4 and 5 in your text; if accepted, production will need this reference to link the reader to the Table.

Author response: We have corrected/ modified in the main document based on the comments

Response to reviewer comments 

Subject: Reviewer comment for Research Article PONE-D-22-34120

Thank you very much for the opportunity given for me to review the manuscript entitled "Magnitude and Determinants of improved household latrine Utilization in Ethiopia: Multilevel Analysis of the Mini Ethiopian Demographic Health Survey (EDHS) 2019."

As mentioned, the authors did a secondary data analysis that used to determine factors associated with an improved latrine Utilization based on the mini–Ethiopian Demographic Health Survey (EDHS) 2019 data set. Given that, I forwarded my comments that I believed which could help authors to improve the quality of their manuscript. 

General comment: 

Overall, despite the known limitation, this study may provide complementary evidence related to latrine Utilization in Ethiopia. I think, the manuscript need language revision and it would be good to refer similar research articles and or tools used for secondary data analysis and follow the writing styles of standard (peer reviewed) published articles. There is mix-up of words and or phrases that are commonly used to report primary studies. For example, phrases such as “This study was conducted in Ethiopia” may give wrong impression and it seems this study was designed to undertake independent study that collect primary research data. Hence, I suggest the authors should revise the document thinking as you are using a secondary data that was collected as part of the national survey. 

Author response: We would like to thank and appreciate the reviewer for providing important comments and suggestions. We have now modified the document based on the comments provided by the reviewer in the main document after reviewing some similar research articles in the area.

Specific comments 

#Introduction: Though the objective of the current study is stated in the abstract, the rationale and aim of the study is not clearly explained in the main document (introduction). Indeed, at the end of the last paragraph the author stated, “The findings of this study will help to increase utilization of improved latrines in the households of Ethiopia”. Can you please justify how? Rather, such type of studies often used to generate further and stronger information that can complement the original survey report. 

Please look at the national EDHS report that was published in May 2021 and articulate the objectives of your study to fil gaps that is not addressed in the report-Perhaps it is still possible to develop testable hypothesis based on your interest of outcome and the main exposure variable. It would be also good to present the conceptual framework that can explain the theoretical and logical relationship of study variables included in this study. 

Author response: We accepted the reviewer comments and suggestions and we have now corrected/ modified in the main document based on the comments

Method: brief description of study settings, design and sampling procedures of the survey might be important but that should not be more detailed. Instead, you can cite the published report and explain the structure and composition of the main dataset related to the main outcome variable and related factor variables used for this study. Similarly, instead of describing in detail the general approaches (methods) used to conduct the survey, it is better to provide enough information on the techniques and procedures on how you extract the study variables and data used to achieve your main objective. 

From my reading (both the manuscript and EDHS survey report), I understood that authors included all eligible samples (households) who were participated in the 2019 mini–Ethiopian Demographic Health Survey. That might be Ok to include all participants, but a systematic sampling (selection) of study units using predefined criteria would be preferred. Meaning, once the maximum power is achieved for the specific objective, the authors should further consider quality of survey data. Not the sample size but such type of dataset requires advanced and sound data handling and management - that is mandatory while using secondary data analysis. 

Similarly, it is not the mathematical formula and or the general assumption (principles) rather authors should clearly explain their own practical analysis techniques and the respective outputs- that used to decide the next procedure. 

I think, definitions of terms should precede data management and analysis- not at the end of research methods. 

Author response: We have corrected/ modified at the main document based on the comments

Results: like that of the method, the descriptive part of your result section reflected the setting and background characteristics of survey participant- it seems the original report. It is not focused as that was expected from secondary data analysis studies. 

It is not clear how and where the X2-test and the corresponding p-vale presented in Table-1 were generated while only the frequency and percentage of one-dimensional variables are presented- which is not cross-tabulated. 

In general, the descriptive findings and the whole results lack the standard criteria used for data visualization (presentation)- perhaps the results (Tabular presentation) reflect the methodological limitation and I felt authors should give more emphasis for data management. For instance, at least those important variables should be recoded or transformed using simple methods and then the reader could grasp the key findings related to this study. 

It is still difficult to understand the findings generated from the multilevel analysis (Modelling). It is not supported with descriptive reports that can provide additional information- If not mistaken I did not see any descriptive findings or estimates that compare the observed frequency related with the main outcome (Improved Vs Unimproved). I suggest either to modify the descriptive findings or add columns to include the observed frequencies [n (%)] in the multilevel analysis (Table 5). 

I think, authors should provide their response and explanation to clarify the above issues and then we can proceed to the next section. 

Author response: We have corrected/ modified at the main document based on the comments. We add descriptive statistical analysis results in the revised document based on the reviewer suggestions.

---

## [Decision Letter · Decision Letter 2]

4 Jul 2023

PONE-D-22-34120R2Magnitude and Determinants of improved household latrine Utilization in Ethiopia: Multilevel Analysis of the Mini Ethiopian Demographic Health Survey (EDHS) 2019PLOS ONE

Dear Dr. Tesfaw,

Thank you for submitting your manuscript to PLOS ONE. After careful consideration, we feel that it has merit but does not fully meet PLOS ONE’s publication criteria as it currently stands. Therefore, we invite you to submit a revised version of the manuscript that addresses the points raised during the review process.

We look forward to receiving your revised manuscript.

Kind regards,

Aiggan Tamene

Academic Editor

PLOS ONE

Journal Requirements:

Reviewers' comments:

Reviewer's Responses to Questions

**Comments to the Author**

1. If the authors have adequately addressed your comments raised in a previous round of review and you feel that this manuscript is now acceptable for publication, you may indicate that here to bypass the “Comments to the Author” section, enter your conflict of interest statement in the “Confidential to Editor” section, and submit your "Accept" recommendation.

Reviewer #1: (No Response)

Reviewer #2: (No Response)

2. Is the manuscript technically sound, and do the data support the conclusions?

Reviewer #1: (No Response)

Reviewer #2: Yes

3. Has the statistical analysis been performed appropriately and rigorously? 

Reviewer #1: (No Response)

Reviewer #2: Yes

4. Have the authors made all data underlying the findings in their manuscript fully available?

Reviewer #1: (No Response)

Reviewer #2: Yes

5. Is the manuscript presented in an intelligible fashion and written in standard English?

Reviewer #1: (No Response)

Reviewer #2: Yes

6. Review Comments to the Author

Reviewer #1: Overall, the authors addressed most of my previous comments and noted a significant improvement on the revised version. Hence, I believed the manuscript can fulfill the required criteria for publication with minor modification/ revision.

Therefore, appreciating the authors commitment, who made further/ major revision and provided satisfactory responses, I would like to remind some of the previous comments, and I think it is better to look the paper once again before production/publication. Hope, you can still improve the language and other editorial/formatting

Method and result section

Do you mean you included all eligible samples (households) and variables from 2019 mini-EDHS? I think, you included all participates (HH) but only selected variable based on your specific objective. If that is the case; it should be stated as “We include all eligible samples (households) who were participated in the 2019 mini–Ethiopian Demographic Health Survey based on predefined selection criteria (study variables) related to the specific objectives.

If you used the full dataset with different/ advanced analytical approaches, that is ok and you clearly described the statistical analysis procedure- though you can still make further data management such as creating new variables or recoding old variables.

• For instance, Number of households member sharing toilet can be recoded in to two/three categories (e.g. < 5 and above) and then you can explain the procedure within data management section.

• In table- 2: remove Type of toilet facility from the last row as it is a duplication of the second variable with merged categories. If interested to present findings for each type of latrine, you can illustrate using separate graphs or chart

• As to my knowledge, simple statistical tests such as X2 and T- test are used to assess or determine the relationship or association between two or more variables – like that of table-4. While there is only one one-dimensional variables in Table- 1. Can you please justify the importance of are X2 - test and the corresponding P- value? If not better to delete the last two columns in table-1

Reviewer #2: (No Response)

7. PLOS authors have the option to publish the peer review history of their article (what does this mean?). If published, this will include your full peer review and any attached files.

Reviewer #1: No

Reviewer #2: **Yes: **Huyen Thi Thanh Dang

---

## [Author Response · Author response to Decision Letter 2]

15 Jul 2023

Thank you very much for all the comments provided regarding our manuscript entitled “Magnitude and Determinants of improved household latrine Utilization in Ethiopia: Multilevel Analysis of the Mini Ethiopian Demographic Health Survey (EDHS) 2019” which are fully accepted and included in the revised version. I have accordingly made necessary revisions on the paper following the comments provided from the reviewer and editor. For your kind consideration, please find a point-by-point response to the comments and a submitted new revised version of the manuscript. All new changes have been highlighted in blue in the main document in order to facilitate review.

I hope that you will find the edits as per your expectation and I look forwards to hear from you

Please kindly find our point-by-point response for the comments provided. 

Author response to reviewer I comments 

Reviewers' comments:

Reviewer #1: Overall, the authors addressed most of my previous comments and noted a significant improvement on the revised version. Hence, I believed the manuscript can fulfill the required criteria for publication with minor modification/ revision.

Therefore, appreciating the author’s commitment, who made further/ major revision and provided satisfactory responses, I would like to remind some of the previous comments, and I think it is better to look the paper once again before production/publication. Hope, you can still improve the language and other editorial/formatting

Method and result section

Do you mean you included all eligible samples (households) and variables from 2019 mini-EDHS? I think, you included all participates (HH) but only selected variable based on your specific objective. If that is the case; it should be stated as “We include all eligible samples (households) who were participated in the 2019 mini–Ethiopian Demographic Health Survey based on predefined selection criteria (study variables) related to the specific objectives.

If you used the full dataset with different/ advanced analytical approaches, that is ok and you clearly described the statistical analysis procedure- though you can still make further data management such as creating new variables or recoding old variables.

Author response: Thank you very much the reviewer for the constructive comments provided. As the reviewer said, in our analysis, we used the “HR data set” based on the guideline of DHS program statistics and the study population were household members who have used latrine in the past five year of the survey. A weighted sample of 8663 individuals were involved in the analysis. Therefore we accepted the comment and we included the suggested statement form the reviewer i.e. “We include all eligible samples (households) who were participated in the 2019 mini–Ethiopian Demographic Health Survey based on predefined selection criteria (study variables) related to the specific objectives”.

• For instance, Number of households member sharing toilet can be recoded in to two/three categories (e.g. < 5 and above) and then you can explain the procedure within data management section.

Author response: We have corrected based on the comment in the main document as suggested by the reviewer. We categorized number of household’s member sharing toilet as < 5 and ≥ 5 household members. 

• In table- 2: remove Type of toilet facility from the last row as it is a duplication of the second variable with merged categories. If interested to present findings for each type of latrine, you can illustrate using separate graphs or chart

Author response: We have corrected based on the comment in the main document as suggested by the reviewer.

• As to my knowledge, simple statistical tests such as X2 and T- test are used to assess or determine the relationship or association between two or more variables – like that of table-4. While there is only one one-dimensional variables in Table- 1. Can you please justify the importance of are X2 - test and the corresponding P- value? If not better to delete the last two columns in table-1

Author response: We have corrected based on the comment in the main document as suggested by the reviewer. As you have said, x2 test is just used for just measuring the association between categorical variables and it will be used as also the first step to run logistic regression model since it is one of the assumptions. But in our study we used it just for showing the association between each individual and community level factors with the outcome variable. Thank you for giving your time to review our paper for improvements 

Author response to reviewer -2

Title: Magnitude and Determinants of improved household latrine Utilization in Ethiopia: Multilevel Analysis of the Mini Ethiopian Demographic Health Survey (EDHS) 2019 

General remarks:

This paper evaluated Magnitude and Determinants of improved household latrine Utilization in Ethiopia, which are interesting information to capture. The manuscript can be accepted for publication after some revisions as below:

Background:

- Please clarify and provide references and clear examples related to “inconsistences between the study’s findings” in the sentence “However, prior studies on the issue are limited and outdated or focused on specific geographical regions in the country, moreover there are inconsistences between the study’s findings” 

Author response: thank you for the reviewer for the constructive comments provided. We have corrected the comments in the main document 

Results:

- When compare with previous studies, the authors should have one or two sentences that discussed the similarity or consistent results of this study and previous studies. How consistent or similar?

Author response: thank you again to the reviewer for the constructive comments provided. We have tried to explain the similarities or the consistence’s between our study and other study findings in the main document 

- Avoid using too much “The odds of”. You should use only one time.

Author response: We have corrected in the main document 

- When discuss the “The variation of the fitted models after Multilevel Analysis”, need to assess which model is the most reliable and can be applied for other similar studies.

Author response: thank you for the reviewer for the comments the provided. Regarding the models, we already discussed about in the methods section. In multilevel analysis due to the hierarchical nature of the data, we have fitted four different models. The first model was the null model which is an empty model developed without individual or community level variables and the second model (Model II) contained the effects of the individual-level variables on the response variable. The third model (Model III) included the influences of the community-level variables on the response variable, and the final model examined the effects of individual-level and community-level variables. The best model to be used in this case is the model which has the lower deviance statistics in relative comparisons between the models. So, a model with a lower deviance (a measure of model fitness) value was considered as the better-fitted model for this study. Therefore, the final model was the best model for our study. 

- Please mention that sex and age are not important factors. Is it consistent with previous studies?

Author response: there are some studies which have found statistical significant association between sex, and age with latrine utilization. These studies are single area studies and most are conducted using a small size. In our study the data was collected at the household level as a national population based survey and age and sex data was taken from the head of the households. 

- Grammar errors: This because these are the big administrative cities in the country and the households in these cities have access to using an improved latrine facility as compared to other regions in the country; Determinate of improved Latrine utilization in Ethiopia, etc.

Author response: we have corrected/edited in the main document 

Conclusion

- English improvement should be conducted.

- There should be included a number found from the result for low utilization of improved latrine.

- In this part, the most recommended model for data analysis should be added.

Author response: Thank you the reviewer for all the constrictive comments provided. We have corrected/edited in the main document. As mentioned in the methods section, the best recommended model for our study is the final model which was developed from both individual and community level factors and it has the lower deviance statistics. 

- The limitation and recommendations for future research and suggestion for more variables during the cross-country census survey for better understanding.

Author response: Thank you the reviewer for all the constrictive comments provided. We have corrected/edited in the main document. We recommend further research to be done at multicounty level survey data i.e. by including Demographic health survey data’s of several countries for making comparisons. In addition further studies should be done also on the full EDHS data set when it is released. 

Conclusion: The paper had good information, however, it needs some revisions and improvement.

 Author response: Thank you for giving your time to review our paper for improvements

---

## [Editor Report · Decision Letter 3]

19 Jul 2023

Magnitude and Determinants of improved household latrine Utilization in Ethiopia: Multilevel Analysis of the Mini Ethiopian Demographic Health Survey (EDHS) 2019

PONE-D-22-34120R3

Dear Dr. Tesfaw,

We’re pleased to inform you that your manuscript has been judged scientifically suitable for publication and will be formally accepted for publication once it meets all outstanding technical requirements.

Kind regards,

Aiggan Tamene

Academic Editor

PLOS ONE
---

## [Editor Report · Acceptance letter]

25 Jul 2023

PONE-D-22-34120R3 

Magnitude and Determinants of improved household latrine Utilization in Ethiopia: Multilevel Analysis of the Mini Ethiopian Demographic Health Survey (EDHS) 2019 

Dear Dr. Tesfaw:

I'm pleased to inform you that your manuscript has been deemed suitable for publication in PLOS ONE. Congratulations! Your manuscript is now with our production department. 

Kind regards, 

on behalf of

Mr Aiggan Tamene 

Academic Editor

PLOS ONE